# The Application of Beta-Tricalcium Phosphate in Implant Dentistry: A Systematic Evaluation of Clinical Studies

**DOI:** 10.3390/ma15020655

**Published:** 2022-01-16

**Authors:** Elisabet Roca-Millan, Enric Jané-Salas, Antonio Marí-Roig, Álvaro Jiménez-Guerra, Iván Ortiz-García, Eugenio Velasco-Ortega, José López-López, Loreto Monsalve-Guil

**Affiliations:** 1Department of Odontostomatology, School of Dentistry, Faculty of Medicine and Health Sciences, University of Barcelona, 08907 Barcelona, Spain; erocamil@gmail.com (E.R.-M.); enjasa19734@gmail.com (E.J.-S.); 2Department of Clinical Sciences, Faculty of Medicine and Health Sciences, University of Barcelona, 08907 Barcelona, Spain; amari@bellvitgehospital.cat; 3Department of Odontostomatology, Faculty of Dentistry, University of Seville, 41013 Seville, Spain; alopajanosas@hotmail.com (Á.J.-G.); ivanortizgarcia1000@hotmail.com (I.O.-G.); evelasco@us.es (E.V.-O.); mmonsalve2@us.es (L.M.-G.)

**Keywords:** beta-tricalcium phosphate, biomaterials, bone substitutes, dental implants, oral implantology

## Abstract

The demand for synthetic graft materials in implant dentistry is rising. This systematic review aims to evaluate the survival rate of dental implants placed simultaneously with bone regeneration procedures using the material β-tricalcium phosphate, one of the most promising synthetic graft materials. The electronic search was conducted in PubMed, Scielo, and the Cochrane Central Register of Controlled Trials. Five randomized clinical trials, one non-randomized controlled clinical trial and four observational studies without control group were include. Implant survival rate and other clinical, radiographic, and histological parameters did not differ from those of implants placed simultaneously with another type of graft material, or placed in blood clots or natural alveolar ridges. Based on the available literature, β-tricalcium phosphate seems to be a promising graft material in implant dentistry. Nevertheless, more randomized clinical trials, with long follow-up periods, preoperative and postoperative CBCT, and histological analysis, are necessary to assess its long-term behavior.

## 1. Introduction

Over the last decades, the increasing demand for implant procedures has promoted the development of multiple bone substitutes from different sources [1,2]. Autogenous bone is the gold standard in bone augmentation, due to its histocompatibility, non-immunogenicity, and great osteogenic, osteoinductive, and osteoconductive capacities [3,4]. However, the increase in morbidity, risk of complications, longer surgical time, and its limited availability often lead to other graft materials being chosen, such as xenografts, allografts, or synthetic materials [4,5,6,7,8].

Allografts come from living or cadaveric human donors, are available in various forms, and are similarly histocompatible. However, due to their processing, they do not have osteogenic capacity, and have lower osteoinductive properties. Moreover, there can be residual immunological risks and a minimal risk of disease transmission [9,10,11].

Xenografts are obtained from non-human species, especially of bovine origin, are biocompatible, and are also available in different formats. They have osteoconductive properties with limited resorptive potential, potentially leading to encapsulation. These grafts have a remote risk of disease transmission and immune response [9,10,11].

Different types of metals, ceramics, and polymers are part of the last group. They are immunologically inert, nontoxic, and the reproducibility of their physical and chemical properties makes the biological reaction to these grafts predictable [5,9]. β-tricalcium phosphate (β-TCP), hydroxyapatite (HA), and their combinations (known as biphasic calcium phosphates (BCP)) [12] are the most studied due to their composition, which is similar to the bone mineral calcium phosphate [5,11,13,14]. They can partially integrate into natural bone tissue, are osteoconductive, and their high affinity for proteins, such as BMPs, can induce stem cell differentiation and growth, and, therefore, new bone formation [9,10]. They can be prepared in granule, block, or putty format, with high porosity to facilitate the penetration and distribution of the vessels [15,16]. HA grafts have slow and limited resorptive potential, so they cannot be replaced entirely by new bone, but can act as volumetric fillers [5,9,11]. In contrast, β-TCP is easily resorbable, and together with its interconnected porous structure, it is rapidly replaced by new bone [5,8,11,17].

Although β-TCP seems to be a promising graft material, the contradictory results obtained in animal studies require further research to clarify the potential of this material in the regeneration of bone defects [16]. The aim of this systematic review is to assess if β-TCP is a good graft material in implant placement surgeries, such as sinus lift procedures, immediate implants, or ridge bone augmentations.

## 2. Materials and Methods

This systematic review was conducted in accordance with the guidelines of the Preferred Reporting Items of Systematic Reviews and Meta-analyses (PRISMA) statement [18].

### 2.1. Focused Questions

Primary question: Is the survival rate of dental implants placed simultaneously with bone regeneration procedures using β-TCP similar to the same procedures using another type of graft material, or similar to dental implants placed without graft material?Secondary Question: Are other clinical, radiographic or histologic parameters similar to the same procedures using another type of graft material, or similar to dental implants placed without using graft material?

### 2.2. PICO Question

P (population): completely or partially edentulous human adults.

I (intervention): implant placement and simultaneous regeneration with β-TCP.

C (comparison): the same procedure using other types of graft or any graft material, or implant placement in natural ridges, or no comparison.

O (outcome): implant survival rate and other clinical, radiographic or histologic parameters.

### 2.3. Eligibility Criteria


*Inclusion criteria:*
Completely or partially edentulous human adults.Placement of dental implants simultaneously with the use of β-TCP.Controlled and non-controlled studies.Randomized clinical trials, controlled clinical trials, clinical trials, clinical studies, comparative studies, observational studies.Studies evaluating implant survival rate.



*Exclusion criteria:*
Case reports and case series.Studies in which implants are placed in second stage after bone regeneration.β-TCP in combination with platelet concentrates or other biomaterials.Immediately loaded dental implants, to avoid the added risk of dental implant failure.


### 2.4. Search Strategy

An electronic literature search for published articles was conducted on 15 October 2021. The consulted databases were PubMed/MEDLINE, Scielo and Cochrane Central Register of Controlled Trials (CENTRAL). Among the results of that search, hand-search was additionally used to identify the articles of interest. The following terms were used for the search: “beta-tricalcium phosphate” AND (“dental implants” OR “implantology” OR “sinus floor elevation” OR “sinus floor augmentation” OR “sinus lift” OR “sinus augmentation” OR ”immediate implants” OR “extraction sockets” OR “GBR” OR “guided bone regeneration” OR “bone augmentation”).

### 2.5. Study Selection

The study selection was carried out by two independent authors (E.R.-M. and E.J.-S.). Duplicates were discarded, and titles and, when necessary, abstracts were read to evaluate their inclusion potential. The full text of the selected studies was read to verify that the inclusion and exclusion criteria were met. A third author (J.L.-L.) was consulted at this stage in order to solve any disagreements.

### 2.6. Data Extraction

The following data were extracted from the included studies: first author, year of publication, country, type of study, number of patients, number of implants, graft material, surgical procedure, follow-up period, survival rate, and number of implant failures. The corresponding authors were contacted when necessary to request missing data.

### 2.7. Quality Assessment and Risk of Bias

Version 2 of the Cochrane Collaboration’s tool for assessing the risk of bias in randomized clinical trials (RoB 2) was implemented to analyze any sources of bias in the following five different domains: bias arising from the randomization process, bias due to deviations from intended interventions, bias due to missing outcome data, bias in the measurement of the outcome, and bias in the selection of the reported result [19].

The Methodological index for non-randomized studies (MINORS) was implemented to evaluate the methodological quality of the non-randomized studies. This tool assesses 8 items in non-comparative studies and 12 items for those with a control group [20].

### 2.8. Data Synthesis and Statistical Analysis

Qualitative and quantitative analyses of all included studies were performed. For the quantitative analysis, the software OpenMeta [Analyst] (Version 1, Brown University, Providence, RI, USA) was used. In particular, a binary random-effects model and the corresponding 95% confidence intervals among the studies were used. The level of significance was set at *p* < 0.05, and the heterogeneity was evaluated based on the I^2^.

## 3. Results

### 3.1. Study Selection

A total of 128 articles were identified through the electronic search. Duplicates were discarded. Titles and, if necessary, abstracts were read to assess the inclusion potential. The full texts of the remaining 12 articles and the 2 selected works identified through hand-searching were evaluated to verify that the inclusion/exclusion criteria were met. Four articles were excluded for the following reasons: implants were placed in second stage [21,22], implants placed in stages one and two were evaluated together [23], and one was a repeated study with a shorter follow-up [24]. Finally, a total of 10 studies were included in the qualitative analysis [25,26,27,28,29,30,31,32,33,34] (Figure 1).

### 3.2. Study Methods and Characteristics

The included studies were published between 2005 and 2021 (Table 1 and Table 2). Five of them were RCTs [26,27,28,29,30], and, in one case, the design was a split-mouth study [29]. One study was a non-randomized controlled clinical trial [25]. The others were observational studies without a control group [31,32,33,34], four of which were prospective [32,33,34] and one of which was retrospective [31].

In all cases, β-TCP (Table 3) was used simultaneously with implant placement in different surgical procedures. In studies with a control group or other test groups, β-TCP was compared to the natural alveolar ridge [25], the blood clot [26,28], deproteinized bovine bone (DBB) [27,28,30], a combination of β-TCP and DBB [28], and platelet-rich plasma (PRP) [29].

In five of the studies, the performed surgical procedure was lateral maxillary sinus augmentation (LMSA) [25,31,32,33,34]. In two of those studies, the procedure was transcrestal sinus lift (TSL) [27,28], and in another one, depending on the residual crest, the procedure was LMSA or TSL [29]. In one study, the procedure performed was immediate implant [26], and the last one performed horizontal bone augmentation procedures [30]. All of them evaluated at least the implant survival rate.

In three studies, a chlorhexidine rinse was performed before surgery [26,28,31]. In most cases, postoperative instructions included antibiotics and non-steroidal anti-inflammatory drugs [25,26,28,30,31,32,33,34]. Only in one study, postoperative antibiotics were not prescribed [27]. Except for two studies (in which it was not specified) [26,28], a chlorhexidine rinse or gel was used for several days after the surgery in all cases [25,27,30,31,32,33,34]. One RCT does not give any information about postoperative instructions [29].

The follow-up period was between 6 months [29] and 146 months [34] after surgery. A total of 741 implants were placed in 405 patients simultaneously with bone regeneration procedures using β-TCP. One study did not specify the number of implants placed [29].

### 3.3. Quality Assessment and Risk of Bias

Figure 2 shows the risk of bias of the included RCTs assessed using RoB 2 [19]. Three had a low overall risk of bias [27,28,30], another had a high risk due to the bias in the measurement of the outcome [29], and the last RCT had some concerns regarding the risk of bias due to deviations from intended intervention [26].

The four non-comparative studies [31,32,33,34] obtained a mean score of 4.5/8 on the MINORS scale. The only non-randomized clinical trial [25] obtained a score of 6/12 on the same scale.

### 3.4. Implant Survival Rate

The implant survival rates of immediate implants (6 months after loading) [26], implants with horizontal bone augmentation (36 months after loading) [30], and implants with TSL (6 months after surgery to 24 months after loading) [27,28,29] were 100%. The implant survival rate in one-stage LMSA was from 96.2% to 100% [25,29,31,32,33,34], being lower in the study with the longest follow-up period (from 104 to 146 months) [34].

In the comparative studies [25,26,27,28,29,30], no differences were found regarding the implant survival rate between the β-TCP group and the control group (PRP) [29]. Additionally, no differences were found regarding natural alveolar ridge [25], blood clot [26,28] or DBB [27,30], or other test groups [28] (DBB and β-TCP+DBB).

#### Meta-Analysis

The survival rate of all the implants placed simultaneously with β-TCP, independently of the surgical procedure, was 98.9% (95% CI: 0.979, 0.999; *p* < 0.001), with no heterogeneity of the included studies (I^2^ = 23.31%, *p* = 0.236) [25,26,27,28,30,31,32,33,34]. One study was not taken into account in this meta-analysis, as it did not specify the number of implants placed, only the survival rate, which was 100% [29] (Figure 3). The implant survival rate in one-stage LMSA using β-TCP was 98.7% (95% CI: 0.972, 1.002; *p* < 0.001), with no significant heterogeneity (I^2^ = 56.242%, *p* = 0.058) [25,31,32,33,34] (Figure 4).

### 3.5. Radiographic Parameters

The bone density around dental implants was evaluated in an RCT, which compared immediate implants, in which the gap was filled with β-TCP, with immediate implants, in which the gap was not filled with any graft material [26]. A statistically significant increase in bone density from 3 to 6 months after loading was only found in the β-TCP group. Additionally, at 6 months after loading, the bone density was significantly different between the groups.

Two RCTs in which TSL was performed analyzed volumetric bone changes around dental implants during the follow-up period, which was 6 months after surgery and 12/24 months post loading, respectively [27,28]. Both of them found that in the β-TCP group, the loss of bone volume was statistically significant over time [27,28], being much more accentuated in the first year after loading, and stabilizing in the second year [28]. Even so, at 2 years after loading, the DBB group had the greatest loss of bone volume (66.34%), followed by the β-TCP group (61.44%), the group with any graft material/coagulum (53.02%), and, finally, the β-TCP+DBB group (33.47%) [28].

In another study, implant placement with simultaneous horizontal bone augmentation with DBB or β-TCP was performed [30]. After three years post loading, no significant differences related to radiographic peri-implant marginal bone loss were observed between the groups.

### 3.6. Clinical Parameters

No statistically significant differences were found related to implant stability immediately after surgery, or over time, in TSL procedures between the groups (β-TCP, DBB, β-TCP+DBB, or blood clot) [28].

The visual analogue scale (VAS), for functional and aesthetic satisfaction, and pink aesthetic score (PES) did not show significant differences in one-stage horizontal bone augmentation procedures between the β-TCP and DBB groups [30].

### 3.7. Histological Parameters

In a prospective study in which β-TCP was used as the grafting material in LMSA, six patients underwent a lateral biopsy 6 months after surgery, and another patient at one year after surgery. Histomorphometric analysis was performed to calculate the ratio of new bone and residual β-TCP, being 11.7% ± 3.0% and 33.2% ± 7.0% at 6 months, respectively, and 34.20% and 6.9% at 1 year, respectively [32].

## 4. Discussion

Although there are few studies on the placement of dental implants simultaneously with different regeneration procedures using β-TCP, the results seem to indicate that the survival rate of these implants is similar to that of implants placed together with other graft materials, or placed in natural alveolar ridges.

Radiographic data indicate that during the first year, the β-TCP undergoes significant reabsorption, which is greater than that of other materials, such as DBB. However, after this period, its reabsorption stabilizes, becoming, after a few years, similar to the reabsorption suffered by other graft materials.

At the clinical level, there appear to be no differences in terms of stability or aesthetic or functional parameters when using β-TCP compared to other graft materials. Histologically, any selected study compared β-TCP with other materials, but in the only one in which a histological analysis was performed, a considerable increase was observed in the ratio of new bone formed, and a significant decrease was observed in the ratio of residual graft formed in 6 months to a year [32].

An RCT comparing the histological differences in horizontal bone augmentation simultaneously to implant placement using β-TCP or DBB concluded that no significant differences were evident at 6 months post surgery [35]. In another split-mouth clinical trial, the formation of new bone in maxillary sinuses in which β-TCP was grafted was compared to sinuses regenerated with autologous bone at 6 months, and no statistical differences were found regarding the new bone density (32.4% ± 10.9% and 34.7% ± 11.9%, respectively) [36]. However, in this study, the biodegradability of the graft was found to be statistically slower in the β-TCP group [36]. Similarly, another RCT, similar to the previous study, also did not observe significant differences between the β-TCP and autologous bone groups at 6 months [37].

No other systematic review has been found to refer to the use of β-TCP in oral implantology, though there is a recent systematic review and meta-analysis published on the use of this graft material in the regeneration of intrabony periodontal defects. This review concludes that β-TCP is a promising material for use in this type of bone defect, obtaining comparable results to other bone graft materials. They also observed that superior outcomes were obtained when this material was combined with growth factors [38].

Another review about the use of β-TCP in maxillofacial and pre-implant surgery concluded that synthetic biomaterials, such as β-TCP, available in the form of granules or compact slabs, guarantee an optimal reconstruction and allow implant placement in the second stage [39].

The main disadvantage of this graft material is its poor mechanical strength [17], which is why it tends to be more often used in contained defects than in areas subject to load. For this reason, in the last years, biphasic calcium phosphates with different HA/β-TCP ratios have been developed, improving its mechanical properties and maintaining a good reabsorption capacity. Even so, the present review includes an RCT in which β-TCP was used in horizontal bone augmentation procedures, and no differences were observed in terms of implant survival compared to the group grafted with DBB [30]. These results are supported by those of an observational study in which, apart from regenerating the atrophic alveolar ridges with β-TCP, the implants were immediately loaded [40].

The present systematic review has several limitations, such as the limited number of selected studies, their heterogeneity regarding the type of study and surgical procedure, and the short follow-up period of some of the studies. In most of the included studies, the surgical procedure was LMSA or TSL. Only one study analyzed immediate implants, and another analyzed implants with simultaneous horizontal bone augmentation, so the results of the use of β-TCP in these last two techniques have little strength.

Beta-tricalcium phosphate and its possible mixtures have a promising future in implant dentistry. More RCTs with long follow-up periods, preoperative and postoperative CBCT, and histological analysis are necessary to really understand how this graft material behaves over time, compared to other commonly used grafts.

## 5. Conclusions

Based on the available scientific literature, the implant survival rates and clinical, radiographic and histological parameters of implants placed simultaneously with bone regeneration procedures using β-TCP seem comparable to those obtained using other graft materials or blood clots, or even to those obtained when the implants are placed in natural alveolar ridges. However, due to the small number of included studies and their heterogeneity, more randomized and multi-centric trials are necessary.

## Figures and Tables

**Figure 1 materials-15-00655-f001:**
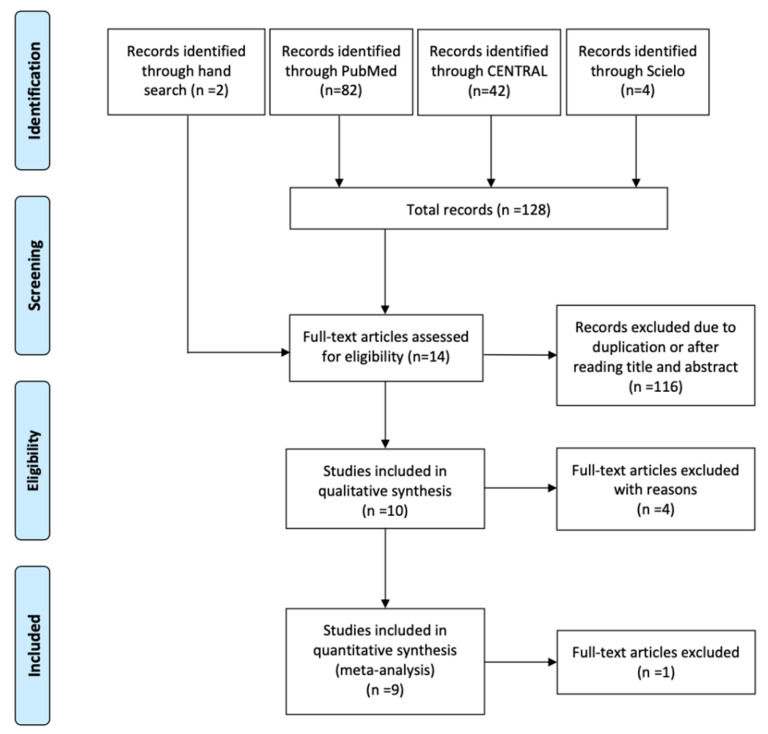
Preferred Reporting Items for Systematic Reviews and Meta-Analyses (PRISMA) flow diagram of selection process.

**Figure 2 materials-15-00655-f002:**
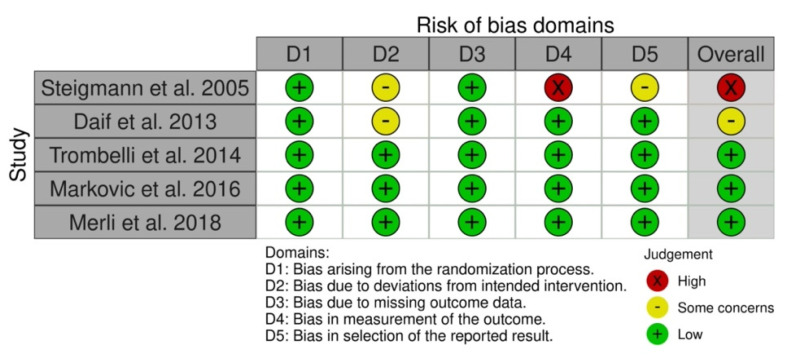
Risk of bias within the included RCTs.

**Figure 3 materials-15-00655-f003:**
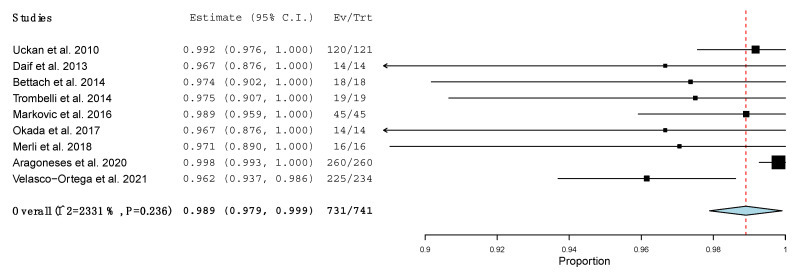
Forest plot of the implant survival rate independently of the surgical procedure.

**Figure 4 materials-15-00655-f004:**
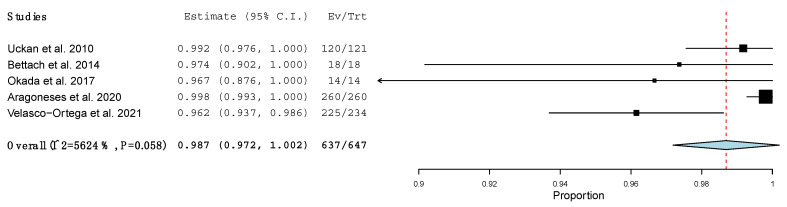
Forest plot of the implant survival rate in one-stage LMSA.

**Table 1 materials-15-00655-t001:** Summary of the included comparative studies. Abbreviations: β-TCP, beta-tricalcium phosphate; CG, control group; CT, clinical trial; DBB, deproteinized bovine bone; HBA, horizontal bone augmentation; LMSA, lateral maxillary sinus augmentation; PRP, platelet-rich plasma; RCT, randomized clinical trial; TG, test group; TSL, transcrestal sinus lift.

Author	Country	Type of Study	N Control		N Test		Control Material	Test Material	Procedure	Follow-up (months)	Survival Rate (Implant Failures)
			Patients	Implants	Patients	Implants					
Steigmann et al., 2005 [29]	USA	Split mouth RCT	20	Not specified	20	Not Specified	PRP	β-TCP	LMSA or TSL	6	100% (0)
Uckan et al., 2010 [25]	Turkey	CT	65	136	62	121	Alveolar ridge	β-TCP	LMSA	CG 32.3; TG 29.8	CG 99.26% (1); TG 99.17% (1)
Daif et al., 2013 [26]	Egypt	RCT	14	14	14	14	Blood clot	β-TCP	Immediate implants	6 after loading	100% (0)
Trombelli et al., 2014 [27]	Italy	RCT	19	19	19	19	DBB	β-TCP	TSL	6	100% (0)
Markovic et al., 2016 [28]	Serbia	RCT	45	45	45	135 (45 every test group)	Blood clot	T1 β-TCP; T2 DBB; T3 β-TCP+DBB	TSL	24 after loading	100% (0)
Merli et al., 2018 [30]	Italy	RCT	18	23	14	16	DBB	β-TCP	HBA	36 after loading	100% (0)

**Table 2 materials-15-00655-t002:** Summary of the included non-comparative studies. Abbreviations: LMSA, lateral maxillary sinus augmentation.

Author	Country	Type of Study	N		Procedure	Follow-up (months)	Survival Rate (Implant Failures)
			Patients	Implants			
Bettach et al., 2014 [31]	France	Retrospective study	4	18	LMSA	22–52	100% (0)
Okada et al., 2017 [32]	Japan	Prospective study	7	14	LMSA	37–46	100% (0)
Aragoneses et al., 2020 [33]	Dominican Republic	Cross-sectional study	119	260	LMSA	6	100% (0)
Velasco-Ortega et al., 2021 [34]	Spain	Prospective study	101	234	LMSA	104–146	96.2% (9)

**Table 3 materials-15-00655-t003:** Most common commercialized β-TCP products (in Spain). Abbreviations: β-TCP, beta-tricalcium phosphate; CDHA, calcium-deficient hydroxyapatite; HA, hydroxyapatite.

Product	Company
Adbone^®^ TCP	Medbone Biomaterials, Lisboa, Portugal
Bonegraft^®^	Bonegraft biomaterials, Turkey
Cerasorb-Curasan^®^	Ancladen, Barcelona, Spain
Iceberg™ TCP	Global Medical Implants, Madrid, Spain
IngeniOs^®^	Zimmer Biomet, Indiana, USA
KeraOs^®^	Keramat, Coruña, Spain
MimetikOss^®^ (20% β-TCP and 80% CDHA)	Mimetis Biomaterials, Barcelona, Spain
OSTEOwelt^®^	Biolot Medical, Turkey
Osteoblast^®^	Galimplant, Sarria, Spain
Powerbone^®^	Medical Expo Bonegraft Biomaterials, Madrid, Spain
R.T.R. Fosfato tricálcico Septodent^®^	Broquer dental , Barcelona, Spain // Contidental, Barcelona, Spain
Straumann^®^ BoneCeramic™	Manohay Dental SA, Alcobendas, Spain
Suprabone TCP^®^	BMT Group, Madrid, Spain-Turkey
SynMax^®^ (40% β-TCP and 60% (hydroxyapatite)	BioHorizons, Madrid, Spain
Trioss^®^	Dilesa, Paterna, Spain
4MATRIX^+®^ (40% β-TCP, 60% HA and hydrogel)	MIS Implants Technologies Ltd, Israel

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
