# Peer review of "The Application of Beta-Tricalcium Phosphate in Implant Dentistry: A Systematic Evaluation of Clinical Studies"

_materials, 2022, doi:10.3390/ma15020655_

Round 1

Reviewer 1 Report

The paper presents a systematic review which is about β-tricalcium phosphate. The innovation of this paper is to evaluate the survival rate of dental implants placed simultaneously with bone regeneration procedures using this material.  It is a topic of interest in researchers in the related areas, but the paper needs some improvement before acceptance for publication. My detailed comments are as follows:

  1. The number of references cited in the article is so small that the degree of credibility is compromised.
  2. The formatting of Tables 1 and 2 in the article is not appropriate and should be revised.
  3. Maybe you could fill in the introductory section to add more of the advantages of beta-tricalcium phosphate to grab people's attention.
  4. Perhaps the reasons for your criteria for screening articles should be explained fully.
  5. It is noted that your manuscript needs careful editing by someone with expertise in technical English editing paying particular attention to English grammar, spelling, and sentence structure so that the goals and results of the study are clear to the reader.

Author Response

Dear Professor,

Thank you very much for considering our paper to be published in your prestigious jorunal.

Below, we answer all the questions that the reviewers indicated.

Use of beta-tricalcium phosphate in implant dentistry: a systematic review (materials-1489842).

Response to Reviewer #1 Comments

The paper presents a systematic review which is about β-tricalcium phosphate. The innovation of this paper is to evaluate the survival rate of dental implants placed simultaneously with bone regeneration procedures using this material.  It is a topic of interest in researchers in the related areas, but the paper needs some improvement before acceptance for publication. My detailed comments are as follows:

Point 1: The number of references cited in the article is so small that the degree of credibility is compromised.

Response 1: We have added some references in the introduction section to explain the advantages and disadvantages of each type of graft material. New references:

  1. Haugen, H.J.; Lyngstadaas, S.P.; Rossi, F,; Perale, G. Bone grafts: which is the ideal biomaterial? J Clin Periodontol. 2019;46:92-102.
  2. Ebrahimi, M.; Botelho, M.G.; Dorozhkin, S.V. Biphasic calcium phosphates bioceramics (HA/TCP): Concept, physicochemical properties and the impact of standardization of study protocols in biomaterials research. Mater Sci Eng C Mater Biol Appl. 2017; 71:1293-1312.
  3. Tavoni, M.; Dapporto, M.; Tampieri, A.; Sprio, S. Bioactive Calcium Phosphate-Based Composites for Bone regeneration. J Compos Sci. 2021; 5(9):227.

Lines [32-64]:

“In the last decades, the increasing demand for implant procedures has promoted the development of multiple bone substitutes from different sources [1,2]. Although autogenous bone is still the gold standard in bone augmentation due to its histocompatibility, non-immunogenicity, and great osteogenic, osteoinductive and osteoconductive capacities [3,4], the increase in morbidity and risk of complications, the longer surgical time and its limited availability, often lead to choosing other graft materials such as xenografts, allografts or synthetic materials [4–8].

Allografts come from living or cadaveric human donors, are available in various forms and are similarly histocompatible, but due to their processing they do not have the osteogenic capacity and have lower osteoinductive properties. There can be residual immunological risks and minimal risk of disease transmission [9–11].

Xenografts are obtained from non-human species, especially of bovine origin, are biocompatible and also available in different formats. They have osteoconductive properties with limited resorptive potential, which can lead to encapsulation. These grafts have a remote risk of disease transmission and immune response [9–11].

Different types of metals, ceramics and polymers are part of the last group. They are immunologically inert, nontoxic and the reproducibility of their physical and chemical properties makes the biological reaction to these grafts predictable [5,9].β-tricalcium phosphate (β-TCP), hydroxyapatite (HA) and their combinations, known as biphasic calcium phosphates (BCP) [12], are the most studied due to its composition similar to the bone mineral calcium phosphate [5,11,13,14]. They can partially integrate into natural bone tissue, are osteoconductive, and their high affinity for proteins such as BMPs can induce stem cell differentiation and growth, and therefore new bone formation [9,10]. They can be prepared in granule, block or putty format, with high porosity to facilitate the penetration and distribution of the vessels [15,16]. HA grafts have slow and limited resorptive potential so they cannot be completely replaced by new bone, but can act as volumetric filler [5,9,11]. Contrary, β-TCP is easily resorbable and together with its interconnected porous structure, it is rapidly replaced by new bone [5,8,11,17].”

Point 2: The formatting of Tables 1 and 2 in the article is not appropriate and should be revised.

Response 2: We appreciate your comment but the tables have been arranged in this format for a better understanding and interpretation of the data. If the tables are divided to place them in a vertical way, their interpretation will bee more complex by the readers.

Point 3: Maybe you could fill in the introductory section to add more of the advantages of beta-tricalcium phosphate to grab people's attention.

Response 3: We have added advantages of beta-tricalcium phosphate in the introductory section. Lines [50-64]:

“Different types of metals, ceramics and polymers are part of the last group. They are immunologically inert, nontoxic and the reproducibility of their physical and chemical properties makes the biological reaction to these grafts predictable [5,9].β-tricalcium phosphate (β-TCP), hydroxyapatite (HA) and their combinations, known as biphasic calcium phosphates (BCP) [12], are the most studied due to its composition similar to the bone mineral calcium phosphate [5,11,13,14]. They can partially integrate into natural bone tissue, are osteoconductive, and their high affinity for proteins such as BMPs can induce stem cell differentiation and growth, and therefore new bone formation [9,10]. They can be prepared in granule, block or putty format, with high porosity to facilitate the penetration and distribution of the vessels [15,16]. HA grafts have slow and limited resorptive potential so they cannot be completely replaced by new bone, but can act as volumetric filler [5,9,11]. Contrary, β-TCP is easily resorbable and together with its interconnected porous structure, it is rapidly replaced by new bone [5,8,11,17].”

Point 4: Perhaps the reasons for your criteria for screening articles should be explained fully.

Response 4: Regarding the two exclusion criteria that may generate doubts:

- Immediately loaded dental implants: we have added the reason of the criteria. Lines [105-106]: “Immediately loaded dental implants, to avoid an added risk of dental implant failure.”

- β-TCP in combination with platelet concentrates: in this case we are referring to any combination of β-TCP with other biomaterials, which may generate bias in the results. We had only written platelet concentrates because it is the most common combination, but we have corrected it as follows: “β-TCP in combination with platelet concentrates or other biomaterials.” [Lines 103-104].

Point 5: It is noted that your manuscript needs careful editing by someone with expertise in technical English editing paying particular attention to English grammar, spelling, and sentence structure so that the goals and results of the study are clear to the reader.

Response 5: The manuscript has been reviewed by a native English speaker. Errors have been corrected throughout the manuscript.

Reviewer 2 Report

The following question mus be clarified before final decision:

  1. the introduction is rather too short and should include more information. Also, the paragraphs are relatively too short and the last paragraph should be merged with the one before that.

2. what do you mean by or no comparison in the PICO question segment (in the C part)?

3. why case series with proper and sufficiently long follow ups were not included?

4. the result section could include informations with the following titles:

types of interventions

presurgical preparations

surgical approach

post-surgical management

comparison groups

and then the measured outcomes mentioned as 3.4 to 3.7 subtitles.

5. the systematic section and meta-analysis should be divided and the related tables for meta-analysis should be added.

6. what is the difference of this study and the 2021 study named" A systematic review on the effect of inorganic surface coatings
in large animal models and meta-analysis on tricalcium
phosphate and hydroxyapatite on periimplant bone formation"

and why wasn't it included in your study?

Reviewer 3 Report

Title: Should be "The application of beta-tricalcium phosphate in implant dentistry: A systematic evaluation of clinical studies"

Introduction: Good

M&M: Well written

RESULTS: Well written and presented. 

Insert a table with the available beta-tricalcium phosphate commercials in order for clinicians to know what products are available in their country.

Discussion: You should write more about the comparison with beta-tricalcium phosphate and other types of bone grafts; Mention advantages and disadvantages of each category. Here I suggest some sources of inspiration (Histol Histopathol. 10.14670/HH-18-108; Drug Metab Rev. 10.1080/03602532.2019.1610767; J Clin Periodontol. 10.1111/jcpe.13058)

Conclusion: Well written.

Please correct grammar and phrases.

Round 2

Reviewer 2 Report

Please check for English errors and plagiarism.
